# Gene Flow and Diversification in *Himalopsyche martynovi* Species Complex (Trichoptera: Rhyacophilidae) in the Hengduan Mountains

**DOI:** 10.3390/biology10080816

**Published:** 2021-08-23

**Authors:** Xi-Ling Deng, Adrien Favre, Emily Moriarty Lemmon, Alan R. Lemmon, Steffen U. Pauls

**Affiliations:** 1Entomology III, Senckenberg Research Institute and Natural History Museum, Senckenberganlage 25, 60325 Frankfurt am Main, Germany; adrien.favre@senckenberg.de; 2Institute of Insect Biotechnology, Justus-Liebig-University Gießen, Heinrich-Buff-Ring 26, 35392 Gießen, Germany; 3LOEWE Centre for Translational Biodiversity Genomics (LOEWE-TBG), Senckenberganlage 25, 60325 Frankfurt am Main, Germany; 4Department of Scientific Computing, 400 Dirac Science Library, Florida State University, Tallahassee, FL 32306-4102, USA; chorusfrog@bio.fsu.edu (E.M.L.); alemmon@fsu.edu (A.R.L.); 5Department of Biological Science, Florida State University, 89 Chieftan Way, Tallahassee, FL 32306-4295, USA

**Keywords:** Hengduan Mountains, *Himalopsyche martynovi*, gene flow, morphology, phylogeny, speciation, target enrichment

## Abstract

**Simple Summary:**

*Himalopsyche* is a group of aquatic insects endemic to the Hengduan Mountains, of which species are usually easily identifiable based on male genitalia, except for a few morphologically variable groups including the *Himalopsyche martynovi* complex. In order to clarify species boundaries within this complex, we investigated its evolutionary history (phylogenetics and gene flow analyses) using a large genomic dataset (~500,000 sites). We found three clades in the *Himalopsyche martynovi* complex, one of which being very variable morphologically while being involved in gene flow with other related lineages. When interpreted in the light of past geological, climatic and palaeohydrological changes, our study suggests that biological novelty—here, trait variation and recombination—may have been acquired via hybridization and represent a source of mountain biodiversity.

**Abstract:**

The Hengduan Mountains are one of the most species-rich mountainous areas in the world. The origin and evolution of such a remarkable biodiversity are likely to be associated with geological or climatic dynamics, as well as taxon-specific biotic processes (e.g., hybridization, polyploidization, etc.). Here, we investigate the mechanisms fostering the diversification of the endemic *Himalopsyche martynovi* complex, a poorly known group of aquatic insects. We used multiple allelic datasets generated from 691 AHE loci to reconstruct species and RaxML phylogenetic trees. We selected the most reliable phylogenetic tree to perform network and gene flow analyses. The phylogenetic reconstructions and network analysis identified three clades, including *H. epikur*, *H. martynovi* sensu stricto and *H.* cf. *martynovi*. *Himalopsyche martynovi* sensu stricto and *H.* cf. *martynovi* present an intermediate morphology between *H. epikur* and *H. viteceki,* the closest known relative to the *H. martynovi*-complex. The gene flow analysis revealed extensive gene flow among these lineages. Our results suggest that *H. viteceki* and *H. epikur* are likely to have contributed to the evolution of *H. martynovi* sensu stricto and *H.* cf. *martynovi* via gene flow, and thus, our study provides insights in the diversification process of a lesser-known ecological group, and hints at the potential role of gene flow in the emergence of biological novelty in the Hengduan Mountains.

## 1. Introduction

The distribution of species diversity is globally uneven [1,2,3], and areas with an exceptional concentration of species are often qualified as ‘biodiversity hotspots’ depending on their current degradation and need for conservation [4,5]. One of the most outstanding mountainous hotspots of diversity is located in the Hengduan Mountains (Hengduan Mts), Southwest China, a region flanking the Qinghai–Tibetan Plateau to the East [5,6]. There, the process of cataloguing life is still ongoing, and a considerable number of species are probably still unknown. Two of the challenges encountered by taxonomists in describing this diversity are detecting cryptic species [7,8], and identifying species boundaries in the face of gene flow within species complexes [9]. To address these issues, genomic data have become increasingly popular because they allow detecting intra- and interspecific gene flow with more accuracy, and thus shed light on when and how boundaries formed among diverging populations and species.

The origin and evolution of biodiversity in the region of the Qinghai–Tibet Plateau (QTP) (i.e., incl. the Hengduan Mts) is an intricate process, involving multiple local and global changes, resulting in a remarkable accumulation of species. In the literature, the diversification of most taxa is attributed to abiotic changes, either climatic or geological [10], and indeed such environmental modifications could act as dynamic drivers for the speciation process, for instance by modifying geographic barriers. As suggested in the “mountain-geobiodiversity hypothesis” [11], patterns of increased diversification in mountain areas (and specifically in the region of the QTP) are associated with three boundary conditions, including a full elevational zonation (i), a high ruggedness of the terrain providing environmental gradients (ii), as well as climate oscillations which facilitate mountain systems to act as “species-pumps”, i.e., they serve as the background for repeated pulses of elevational migration leading to fragmentation and ultimately speciation followed by species expansions over large time scales [11] (iii). Conditions (i) and (ii) were probably realized early on in the orogeny of the Hengduan Mts, as alpine vegetation evolved during the Oligocene in this region [12]. Many radiations, however strongly overlap temporally with climate oscillations (iii) of the last few million years, at least in plants [10]. During these more recent times, cyclical climatic modifications may have fostered alternate phases of species’ range fragmentation (allopatric differentiation) and secondary contact, upon which gene flow among species or populations with incomplete reproductive isolation may have occurred. Thus, the hypothesis provides a good overview on how dynamic abiotic conditions affect the evolution of mountain biota. In the Hengduan Mts, climate changes and orogenic movements profoundly changed the relative frequency and the distribution of available habitats, affecting species movement and distributions and ultimately speciation and patterns of diversity [13]. Species complexes are common in the Hengduan Mts (e.g., plant [14], mammal [15], fungus [16], caddisflies [17]) and represent good models for studying evolutionary processes and mechanisms of speciation in the context of the mountain-geobiodiversity hypothesis.

*Himalopsyche* is a common taxon in the region of the QTP with strong affinity to high elevation streams and rivers. The genus has its center of diversity in the Hengduan Mts [18]. Currently, the genus encompasses 56 species, mostly described on the basis of stable morphological features of adult male genitalia that display a diverse morphology but very little intraspecific variation. Thus, the morphology of male genitalia is usually sufficient for reliable taxonomic identification [19]. However, for a few species, a larger intraspecific variability of this trait renders species delineation challenging. To cope with these cases, several species complexes were defined, including the *triloba*-complex, the *japonica*-complex, the *excise*-complex and the *martynovi*-complex [17,20,21,22]. The *martynovi*-complex is endemic to the Hengduan Mts and was originally described as the *H. alticola*-*martynovi*-complex by Ross on 1956 [21], and later split as *H. alticola* and *H. martynovi* by Schmid and Botosaneanu [22]. Then, Malicky [23] assigned *H. epikur* to the *martynovi*-complex as a new species based on morphological evidence. Recently, Hjalmarsson [24] attempted to delineate the species of this complex more precisely, using both morphological and genetic approaches. Using five genetic markers, Hjalmarsson [24] showed that *H. epikur* forms a monophyletic group with stable morphological characters and concluded that *H. epikur* is genetically, morphologically and ecologically distinct. However, in that study, *H. martynovi* (as sister clade to *H. epikur*) contained a perplexing morphological variation with some diagnostic traits showing a gradient of intermediate forms between the typical morphologies of the two species. The intraspecific morphological variation within the *H. martynovi* clade is a very unusual case in *Himalopsyche*. Given climate cycles and associated range expansion-regression dynamics of a “species-pump” in the last few million years, some extent of hybridization upon secondary contact may be envisaged as an origin of such intraspecific morphological variation. In fact, Malicky and Pauls [25] found that hybridization between two closely related species of Trichoptera lead to explicitly intermediate morphologies consistent with co-dominant inheritance patterns. Therefore, detecting intraspecific and interspecific gene exchange among species of the *H. martynovi* complex may shed light on how morphological variation arose, whether or not gene flow played a role in its evolution, and help taxonomic delineation among diverging populations and species.

In this study, we thus investigate the different clades of the *H. martynovi*-complex, using phylogenetic reconstructions based upon molecular data from 691 loci captured by anchored hybrid enrichment (AHE; [26]). We aim to compare the resulting phylogenies with the morphological features of male genitalia of adult specimens. In order to improve species delineation, we test (i) whether AHE loci provide enough resolution, (ii) whether gene flow occurred among the different clades and (iii) whether the direction of gene flow may explain the morphological variation observed in the *H. martynovi*-complex.

## 2. Materials and Methods

### 2.1. Taxon Sampling

We included 11 specimens representing five closely related species. In this case, 10 of these samples were adult males and one was a larva. All specimens were determined using Hjalmarsson [24]. *Himalopsyche immodesta* was chosen as outgroup, *H. viteceki*, *H. martynovi* and *H. epikur* were regarded as ingroup species. Specimens were collected in the wild between 2010 to 2014, preserved in >75% ethanol and deposited in the collections of the Senckenberg Research Institute and Natural History Museum. The genitalia of all the adults were cleared in either KOH or lactic acid, then photographed using a DP2 digital camera mounted on an Olympus SZX7 steromicroscope. All specimen and voucher information are shown in Appendix A and Figure 1. *H. epikur* is distributed in southern region of the Hengduan Shan, *H. martynovi* further north-east. The distribution area of *H. viteceki* and the outgroup *H. immodesta* overlap with that of *H. epikur*.

### 2.2. DNA Extraction, Library Preparation, Enrichment and Sequencing

Prior to DNA extraction, wings (in adults), heads and terminal abdominal segments were removed as specimen vouchers. The thoracic or abdominal muscle were then used for the genomic DNA extraction. DNA was extracted with hot sodium hydroxide as described in Truett et al. [27]. Extracted DNA was visualized on an agarose gel to estimate the mean size of the fragments. Concentration of double stranded DNA was fluorometrically estimated using a Qubit assay kit (Thermo Fisher Scientific, Waltham, MA, USA). The DNA pellet was washed 2× with 70% EtOH. Following the wash, the EtOH was discarded and the pellet dried. The pellet was sent dried to the Center for Anchored Phylogenomics at Florida State University (http://www.anchoredphylogeny.com, accessed on 22 April 2016) for quality check and library preparation. The library preparation was conducted by 4 steps: (1) sonicated extracted DNA to a size range of approximately 200–500 bp using a Covaris Ultrasonicator (Covaris, Woburn, MA, USA). (2) ligated common Illumina adapters. (3) indexed with 8 bp (single) indexing adapters. (4) pooled the libraries in equal concentration. In addition, the libraries were enriched using probes designed for Trichoptera (as described in the next section). Afterwards samples were sequenced on one lane of Illumina Hiseq sequencer with a paired-end 150 bp protocol following Lemmon et al. [26] and Prum et al. [28]. The Florida State University College of Medicine Translational lab performed the sequencing.

### 2.3. Probe Design

The probes were designed in collaboration with the Center for Anchored Phylogenomics. Target AHE loci in common among Trichoptera and other Holometabola were identified by scanning 15 trichopteran 1-Kite transcriptomes (see Appendix A for details) for the Lepodoptera AHE loci identified by Breinholt et al. [29]. The transcripts identified were aligned in MAFFT v7.023b [30] by target locus, then trimmed to well aligned regions and finally manually inspection in Geneious R9 [31]. A total of 989 target loci (averaging 232 bp in length) remained after masking/removing regions identified to be repetitive using kmer distribution profiling (see [32] for details). For each of these loci, probes were tiled uniformly across each of the 16 reference sequences at with 4.2× coverage, to produce 57,094 probes. Probes were produced by Agilent as a SureSelect XP kit. The final alignments used for probe tiling, as well as the probe design itself is given as Appendix A.

### 2.4. Raw Data Processing and Assembly

Raw read pairs passing the CASAVA high-chastity filter were merged following Rokyta et al. [33] and library adapters were trimmed. Reads were then assembled to the reference *Rhyacophila fasciata* (probe region sequences) using the quasi-de novo assembly procedure described by Hamilton et al. [32]. Consensus sequences were derived from assembly clusters containing 10 or more reads and ambiguities were called when site patterns could not be explained by a 1% sequencing error. Orthology among consensus sequences was determined using a neighbor-joining approach based on a pairwise distance matrix (see Hamilton et al. 2017 for details). Alleles were then phased following Pyron et al. [34] to produce two sequences per individual.

### 2.5. Alignment and Trimming

Each locus containing two allelic sequences for each individual was aligned using MAFFT v. 7.023b [30] separately. To generate a high-quality dataset, we then performed data filtering for each locus as follows: (1) we removed the columns which represented less than 75% of individuals on two ends of each alignment with trimAl [35]. (2) We identified the randomness section in each alignment with Aliscore and then removed it with Alicut [36]. (3) We discarded loci with a length <400 bp. (4) Finally, we also discarded alignments that contained less than 16 individuals. The remaining 691 loci were then analyzed both in concatenation and individually (Appendix A).

### 2.6. Phylogenetic Analyses

Prior to phylogenetic analyses, we generated four datasets based on the allele alignments after data filtering in order to estimate the phylogenetic accuracy at allelic level. Indeed, the multiple sequence alignment based on alleles is known to be the optimal data type for phylogenetic analyses because it contains additional heterozygous information and represents the smallest units in evolution [37]. These four datasets were: (1) multiple sequence alignments (MSAs) of each locus and a concatenated alignment containing only allele 1 (allele 1 alignment); (2) MSAs of each locus and a concatenated alignment containing only allele 2 (allele 2 alignment); (3) MSAs of each locus and a concatenated alignment containing both alleles (bi-allelic alignment); (4) MSAs of each locus and a concatenated alignment containing merged alleles based on the International Union of Pure and Applied Chemistry (IUPAC) ambiguity codes (merged-allele alignment). Then, we conducted the phylogenetic analyses on these four datasets separately.

We inferred the maximum-likelihood (ML) tree with the concatenated loci in RAxML v8.2.X [38]. The ML tree was generated with 1000 bootstrap replicates and partitioned by each locus. We conducted a model test with the concatenated sequence using jModelTest 2.1.4 [39,40], which identified the GTR+I+G model as the most appropriate (Appendix A). However, according to the opinion of Stamatakis [38], GTRGAMMA is to be preferred over GTR+I+G for small datasets. Thus, we selected GTRGAMMA model instead of GTR+I+G in this case. We used the same approach to construct the ML tree for each single locus but using 100 bootstrap replicates instead of 1000. Since the single locus only contained 2–7% informative sites, these single gene trees showed a highly incongruent phylogenetic pattern. Finally, we generated a summary-based species tree based on the 691 unrooted gene trees using ASTRAL-III [41].

To better visualize the genealogical concordance and discordance, as well as getting an initial assessment of potential gene flow, we conducted a network analysis using SplitsTree4 ver.4.16.2 [42] of 571 gene trees which contains all individuals selected from the 691 trees. We used ConsensusNetwork as the tree transformation and three different thresholds (3%, 5% and 20%) in SplitsTree v4.

### 2.7. Gene Flow Analysis Using ExD_FOIL_

Usually, hybridization is defined as the interbreeding between two species or two genetically distinct lineages [43], and introgression as gene transfer between species or differentiated population by backcrossing, potentially leading to a permanent incorporation of some genes (and traits) of one species into the other [44]. In parallel, gene flow is a collective term that includes all mechanisms resulting in the movement of genes from one population to another [45,46]. In our case, as we deal with a species complex, we will use the terms “introgression” for gene exchange between two well-supported species, and “gene flow” within the species complex itself.

A SNP dataset was called from two allele alignments using SNP-sites [47] (https://github.com/sanger-pathogens/snp-sites, accessed on 12 December 2020) and then used to detect potential introgression among species. To better grasp the speciation mechanisms, we employed ExD_FOIL_ which uses allelic patterns to detect interspecific introgression [48,49] (https://github.com/SheaML/ExDFOIL, accessed on 10 November 2020). ExD_FOIL_ test is an extension of the D_FOIL_ analysis, which we applied to a symmetric five-taxon phylogeny as (((P1, P2), (P3, P4)), O) [48]. Typically, the program is able to detect introgression between lineages, and suggest its direction. It can also estimate whether introgression is direct or indirect, current or ancestral. We used *H. immodesta* as outgroup and the species tree calculated by bi-allelic alignment as the tested topology because of its high reliability. Each allele was regarded as one individual in the approach. In total we tested 1755 suitable combinations for reconstructing introgression. Afterwards the predicted introgression between individuals was summarized as introgression between lineages.

## 3. Results

### 3.1. AHE Target Characteristics

The target locus set contained 989 loci averaging 232 bp per locus (range: 150–1721 bp). Note however that 37 of the loci were missing a *Rhyacophila* sequence as outgroup. Consequently, the maximum number of loci that could be obtained in this study is 952. The total target size was 367,184 sites. Pairwise sequence identity averaged 73.39% (range: 87.2–50.3%). The taxon coverage was very good, with 97% of the loci containing sequences for at least 10 of the 11 species. As a result, the alignments were very complete; only 3.9% of the alignment characters were missing or ambiguous.

### 3.2. Read and Assembly Characteristics

The sequencing effort produced 95 million raw read pairs, averaging 1.6 Gb per sample (Appendix A). Reads mapped to the reference sequence with a 24.6 on-target percentage. On average, 738 loci (with consensus sequences at least 250 bp in length) were obtained per sample. Consensus sequences, constructed from an average of 3446 reads per locus, averaged 630 bp in length. Due to the high coverage per locus, the PCR duplicate rate was moderate (approximately 50%).

### 3.3. Sequence Characteristics after Filtering

After filtering, the alignments contained a total of 509,113 sites including 691 loci, of which 65.8% were identical sites and 4% were missing characters (Appendix A). Individual alignments of each locus ranged from 401 bp to 2293 bp in length, the mean length being 737 bp. Each of these loci were recovered for more than eight specimens included in this study and were used for the subsequent phylogenetic analyses.

### 3.4. Phylogeny

We reconstructed a species tree based on 691 loci, and a RaxML tree based on concatenated loci for four different alignments (allele 1 alignment, allele 2 alignment, bi-allelic alignment and merged-allele alignment). All analyses recovered almost identical topologies regardless of the datasets used. Most nodes were highly supported (BS > 95), especially those forming the backbone of the phylogeny, whereas some uncertainty occurred between species tree and RaxML tree for some datasets. The species relationship of *Himalopsyche* using two approaches based on four allelic procedures were highly consistent, and all phylogenetic trees showed four monophyletic groups: *H. viteceki*, main clade of *H. martynovi*, one individual lineage of *H. martynovi* and *H. epikur*. For convenience, we will refer to the main clades as *H. martynovi* sensu stricto (*H. martynovi* s.s.) and the single individual lineage as *H.* cf. *martynovi*, respectively. *Himalopsyche* cf. *martynovi* was identified as *H. martynovi* based on morphological and molecular evidence in earlier studies [24], but clustered as a sister clade to *H. epikur* in our analyses. Finally, *H. viteceki* was sister to the clade containing *H. martynovi* s.s., *H.* cf. *martynovi* and *H. epikur*.

The coalescent species trees and concatenated RaxML trees were consistent for most of the clades, except for a few shallow nodes in *H. epikur* and *H. martynovi* s.s. when calculated using the allele 1 alignment and the bi-allelic alignment (Figure 2). Notably, the topological structure of *H. martynovi* s.s. were incongruent comparing the species tree with the ML tree using the three alignments which contained only one allele (allele 1 alignment, allele 2 alignment, merged-allelic alignment) (Figure 2A–C), but congruent when using the bi-allelic alignment. In all the species trees calculated by one single allele alignment, the individuals of *H. martynovi* s.s. are clustered as four monophyletic sister lineages with some low support values. All trees showed two pairs of sister branches in all four ML trees and the one Astral tree based on the bi-allelic alignment. In addition, the phylogenies based on bi-alleles’ alignments comprise the most complete data set and showed greatest consistency between species tree and RaxML tree in all the key nodes (densitree in Figure 2D). Hence, we chose the species tree calculated with the bi-allelic alignment for the subsequent gene flow analysis, as these required a stable and accurate topology.

The consensus network analysis showed unambiguous clusters of species (Figure 3). Samples of *H. epikur* formed a cluster with simple branch structure at all the thresholds. *Himalopsyche martynovi* s.s. clustered together and formed more internal reticulation within the clade at a lower threshold. *Himalopsyche* cf. *martynovi* clustered as sister to *H. epikur* at the thresholds of 3% and 5%, but clustered together with *Himalopsyche martynovi* s.s. at the thresholds of 20%. The outgroups and *H. viteceki* were recovered as two independent lineages without any reticulation toward any of the other species regardless of the threshold.

### 3.5. Introgression

The ExD_FOIL_ analyses revealed numerous signals of gene flow among the four taxa (excluding *H. immodesta*, the outgroup) (Figure 4, Appendix A). For example, the analysis detected frequent bidirectional gene flow (counts = 10) between different individuals of *H. epikur* and one subclade of *H. martynovi* s.s.. Introgression is also likely to have occurred between *H. martynovi* s.s. and *H.* cf. *martynovi* (counts = 11). Finally, there were some unidirectional signals (counts = 2) of gene flow from one subclade of *H. martynovi* s.s. into *H.* cf. *martynovi*. In addition, there was one bidirectional gene flow signal between *H. viteceki* and *H.* cf. *martynovi*.

### 3.6. Morphological Sorting

Figure 4 associates the morphology of adult male’s genitalia with the phylogenetic context. We found uniform morphological characteristics within *H. viteceki* and *H. epikur* based upon two and three individuals, respectively. In contrast, the morphology for *H. martynovi* s.s. and *H.* cf. *martynovi*, was more complex and variable within *H. martynovi* s.s. (based upon four individuals). However, these two lineages were not only separated by molecular evidence, but also present distinct morphologies. The main morphological differences between these two lineages were: (i) the incised curving in the middle of the superior appendages in lateral view was slightly incised in some of the individuals of *H. martynovi* s.s. but strongly concave in other individuals of *H. martynovi* s.s. and *H.* cf. *martynovi* (ii) the distal margin of the superior appendages in lateral view, had a protruding shape in *H. martynovi* s.s. and a concave shape in *H.* cf. *martynovi.* We found that individuals of the *H. martynovi*-complex often displayed intermediate morphological features in the superior appendages, thus partly mirroring the phylogenetic relationships. Specifically, *H.* cf. *martynovi* appeared as an intermediate form between *H. martynovi* s.s. and *H. epikur*, and *H.* cf. *martynovi* and *H. martynovi* s.s. had an intermediate morphology between *H. viteceki* and *H. epikur*. In addition, the intraspecific morphological variation of *H. martynovi* s.s. was on par with interspecific morphological variation in the complex.

## 4. Discussion

Species complexes are taxonomically challenging because they are usually characterized by a constellation of morphological traits which are not clade-specific, and often traditional molecular markers do not suffice to resolve phylogenetic relationships convincingly. In this study, we used genomic data associated with morphological investigations in order to clarify species delineation within the *H. martynovi*-complex. Endemic to the Hengduan Mts, this species complex is characterized by an unusually high intraspecific morphological variation and interspecific morphological gradients [24]. We thus investigated whether gene flow may be the source of this morphological variation. Our study is the first to use an advanced molecular methodology (AHE) to investigate a species complex in caddisflies and more generally to link the resulting pattern with gene flow as a mechanism potentially responsible for the morphological ambiguity and biological novelty.

### 4.1. Genomic and Morphological Evidence and Their Taxonomic Implications

In order to test the reliability of phylogenetic reconstructions, we used four different datasets based on different allele sequences, namely alignment including either allele 1 or 2, as well as a bi-allelic and a merged-allelic alignment. Our results on allele phasing are in line with the study of Andermann et al. [37] and reveal that the data set including two alleles generally recovers a more accurate estimate of tree topology in terms of less ambiguity and high consistency in the species tree approaches. This was particularly true for phylogenetic relationships among the study’s focal clades. Moreover, the main tree topologies were consistent across both species trees and ML trees using concatenated alignments within each dataset. Thus, the reduced genomic datasets we generated by AHE appear to improve our understanding of ambiguous taxonomic relationships within this caddisfly species complex as previously shown for other arthropods [29,32,50,51,52].

As expected, we identified three robustly supported (100 BP) main clades, corresponding to *H. martynovi* s.s., *H. epikur* and *H. viteceki*, respectively. However, we uncovered a fourth lineage which we called *H.* cf. *martynovi*. This lineage appears as sister to *H. epikur* in all trees, as well as in the network analyses. *Himalopsyche* cf. *martynovi* might represent another enigmatic species called *H. alticola*, which was originally described as one of three main species of the *H. martynovi*-complex (with *H. martynovi* and *H. epikur*). However, the description of this species is rather incomplete, providing only coarse explanations and a single drawing of its morphology [20,22]. Thus, although there is a morphological resemblance between *H.* cf. *martynovi* and *H. alticola* on the basis of the rudimentary species description, the morphological information available for this latter species is too unreliable to conclusively attribute *H.* cf. *martynovi* to *H. alticola*, especially in the presence of the unusual morphological variation in the *H. martynovi*-complex. Further morphological and molecular evidence would be needed to evaluate the validity of *H. alticola*, in particular from its holotype or at least from its type locality. While the sampling of our study is limited in the number of individuals, but it includes all hitherto known morphological variation and most of the distribution range of this group. Moreover, our study mainly focuses on the origin and intraspecific variation pattern in this species complex, which can be readily assessed with our data.

From our results it is clear that there are at least three lineages in the *H. martynovi* complex even though there is unusually high morphological variation among the specimens of *H. martynovi* s.s. Interestingly, *H.* cf. *martynovi* presents an intermediate morphology between *H. martynovi* s.s. and *H. epikur* (Figure 4). For example, the incised curving in the middle of the superior appendages of *H.* cf. *martynovi* is concave to an intermediate degree compared with specimens of *H. martynovi* s.s. and *H. epikur*. Moreover, the distal margin of the superior appendages in *H.* cf. *martynovi* shows an intermediate form between the protruding shape in *H. martynovi* s.s. and the concave shape in *H. epikur*. The concordance of the molecular and morphological evidence suggests that either a stepwise evolution of traits occurred among these three lineages, or alternatively that intermediate morphologies in *H.* cf. *martynovi* were acquired via gene exchange between or with *H. martynovi* s.s. and *H. epikur*.

### 4.2. Gene Flow and Speciation of H. martynovi Complex

We found that species of the *H.*
*martynovi*-complex, and particularly *H. martynovi* s.s., cannot easily be distinguished morphologically because of the high versatility of trait morphology in the male genitalia, whereas our phylogeny unequivocally recovers three robust lineages in this species complex. In parallel, we find evidence for gene flow or introgression not only within lineages (network: within *H. martynovi* s.s. and *H. epikur*), but also among lineages/species (ExD_foil_ analysis: among the three lineages and a faint introgression between *H. viteceki* and *H. martynovi* complex). This line of evidence, coupled with the intermediate morphology of *H.* cf. *martynovi* and the variable nature of morphological traits in *H. martynovi* s.s. suggests that these lineages may bear the morphological signature of gene flow. This would not be an isolated case in caddisflies, since it has already been shown in Limnephilidae that hybrid males would present an intermediate morphology for their genitalia [25].

In fact, it is increasingly accepted that hybridization, gene flow and introgression may play a supporting role during speciation and diversification, because they foster new gene combinations and the transfer of adaptive genes [53,54,55]. Even though this concept is more accepted for plants, some evidence indicates it is also the case for animals [56,57,58,59,60]. For example, hybridizing lineages of salamanders have been shown to have significantly greater net-diversification rates than non-hybridizing lineages [61]. Our analyses showed that *H. martynovi* s.s. is likely to have been involved in introgression with other lineages, possibly making it the recipient of genes corresponding to contrasting trait values, which today still co-exist in this lineage, explaining its remarkable morphological diversity. However, *H. epikur*, which was one of the known partners involved in gene flow, does not display such morphological variation. One can only assume that this discrepancy could be due to a bidirectional but asymmetric gene flow (predominantly towards *H. martynovi* s.s.), or that selective pressures have eliminated parts of the morphological variation resulting from gene exchange in *H. epikur* and not in *H. martynovi* s.s.

Considering the current distribution ranges of species of the *H. martynovi*-complex and our knowledge on range displacement during climate oscillations for organisms of the Hengduan Mts, we hypothesize that our results match with a general species-pump scenario. For example, it is often reported that species or populations have diverged allopatrically in the northern and southern Hengduan Mts, respectively [15,62,63]. This might have been the case for *H. epikur* (southern Hengduan Mts) and *H. martynovi* s.s. (northern Hengduan Mts), since these two species occur in drainage systems that were historically distinct [64]. Indeed, the upper Yangtze River (the distribution area of *H. epikur*, Figure 1) and the Yalong-Dadu river (the distribution area of *H. martynovi* s.s.) remained separated until ca. 1.3 Ma [64], suggesting a possible minimum divergence time for these two species. After their allopatric divergence, re-organization of drainage systems coupled with climate oscillations may have fostered repeated encounters between these two species, upon which gene flow may have occurred, as is predicted by the “*mountain geo-biodiversity hypothesis*” [11]. We believe it should come as no surprise that *H.* cf. *martynovi*, phylogenetically closely related to *H. epikur* but morphologically more similar to *H. martynovi* s.s., has been found at the southwestern edge of the distribution range of the latter, closest to known populations of the former (see Figure 1). The area between the Dadu and Yalong rivers may thus have acted as a contact zone between the two species. If this were formally verified, our study would be one of the very few to document the role of gene flow as a source of biological novelty under the impulse of a species-pump effect. However, sympatric species such as *H. viteceki* and *H. epikur* do not seem to have a history of gene flow, which would counterbalance our hypothesis. We argue that not only a stricter ecological differentiation may have evolved between these two species (as is often the case in sympatry), or alternatively, that their more ancient divergence promotes more genetically-based incompatibilities today, hence protecting *H. viteceki* and *H. epikur* from hybridization. Admittedly speculative, our hypothesis is nevertheless testable, for example by extending the sampling to represent all secondary drainage systems in the area, and by dating the divergence of the lineages of the *Himalopsyche martynovi* species complex.

## 5. Conclusions

The phylogenetic analyses based on genomic data captured by AHE identified three lineages in the *H. martynovi*-complex, one of them unequivocally identified as *H. epikur*. The other two lineages, which we named *H. martynovi* s.s. and *H.* cf. *martynovi*, were similar to each other and morphologically despite trait variation, and intermediate between *H. viteceki* and *H. epikur*. We showed that gene flow occurred frequently between *H. martynovi* s.s. and *H.* cf. *martynovi*, as well as between *H. martynovi* s.s. and *H. epikur*. We argue that gene flow may in fact be the source of morphological variation in *H. martynovi* s.s., and intermediate morphotypes in *H.* cf. *martynovi*. Climate oscillations and re-arrangement of local drainage systems may have fostered ancient and recent introgression upon secondary contact in this species. Finally, our study shows that in genera where diagnostic traits are considered to be stable and thus excellent for the determination of most species (as the male genitalia in caddisflies), morphological attributes may not be stable in all species. Therefore, an integrative approach such as ours should be taken into consideration for taxonomy as it avoids over-splitting morphologically versatile lineages, especially for taxa characterized by limited diagnostic features [65,66,67,68].

## Figures and Tables

**Figure 1 biology-10-00816-f001:**
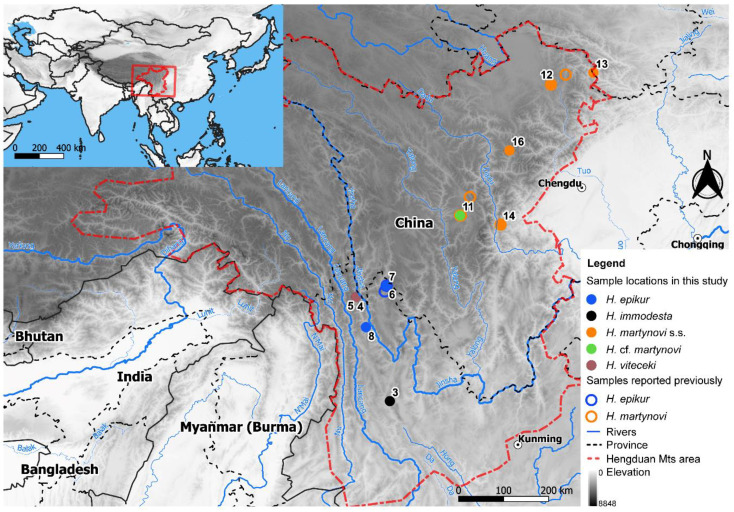
Sample locations of *Himalopsyche* species used in this study (solid dots) and compiled from the literature (hollow dots). *H. martynovi* s.s. is the abbreviation of *H. martynovi* sensu stricto.

**Figure 2 biology-10-00816-f002:**
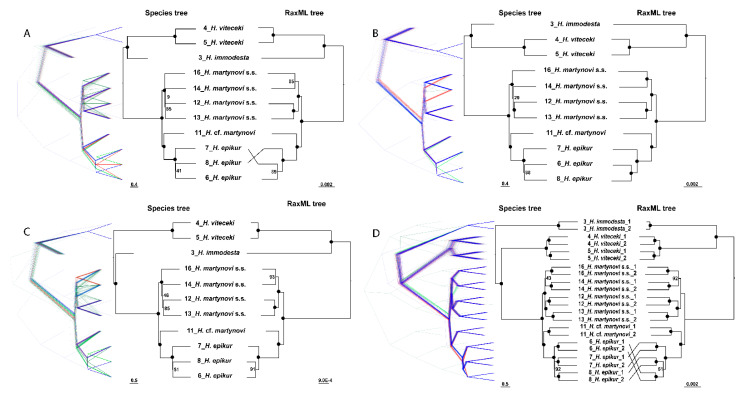
Species tree with 691 loci (left) and RaxML tree with the concatenated loci (right) for *Himalopsyche martynovi* complex based on four datasets used in this study: (**A**) with allele 1; (**B**) with allele 2; (**C**) with the merged allele based on IUPAC code. On each figure, the left part shows a plot of the complete bootstrap species tree distribution using DensiTree (**D**). The black circle on the branch node represents bootstrap scores above 96, otherwise is shown with numbers. *H. martynovi* s.s. is the abbreviation of *H. martynovi* sensu stricto.

**Figure 3 biology-10-00816-f003:**
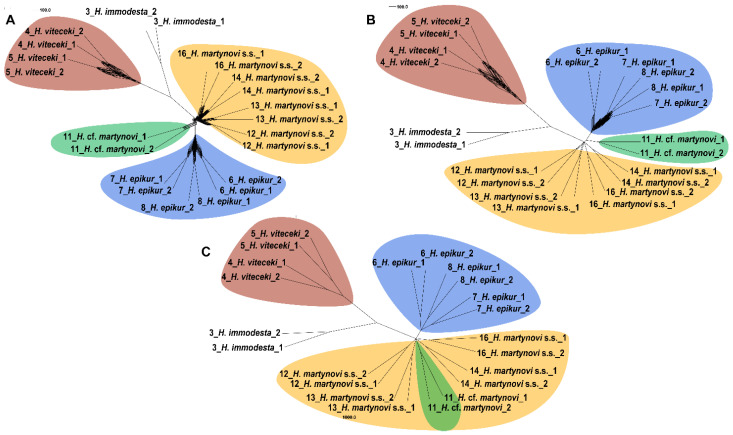
Network of *Himalopsyche martynovi* complex based on all the RaxML gene tree using SplitsTree. The thresholds were set to 3% (**A**), 5% (**B**) and 20% (**C**), respectively. *H. martynovi s.s.* is the abbreviation of *H. martynovi* sensu stricto.

**Figure 4 biology-10-00816-f004:**
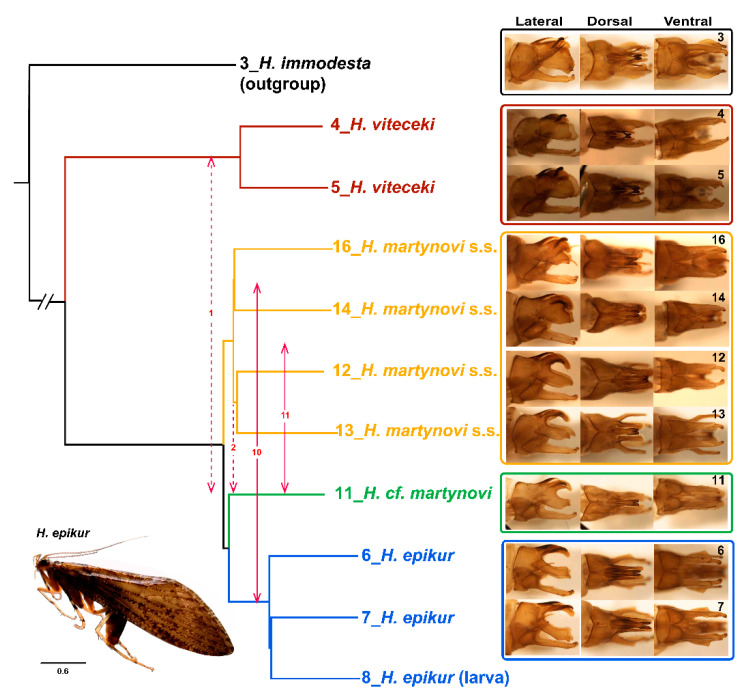
Gene flow and morphological variation of the genitals among the *Himalopsyche* species showing on the species tree. The species tree is calculated by ASTRAL-III based on 691 gene trees constructed with maximum likelihood. Different colors show the phylogenetic position among the same species. Gene flow are calculated by ExoDfoil. Two-way arrows depict bidirectional gene flow signal, one-way arrows depict unidirectional gene flow signal, numbers inside arrows represent the frequency of significant gene flow signals. Different thickness of the solid arrows shows different gene flow frequency, dashed red lines indicate low gene flow frequency. *H. martynovi s.s.* is the abbreviation of *H. martynovi* sensu stricto. Numbers on the images are corresponding with the sample ID.

## Data Availability

The raw data of Anchored hybrid enrichment in this study has been deposited at the NCBI SRA database under the Bioproject ID PRJNA744478 (accessed on 7 July 2021).

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
