# Peer review of "Gene Flow and Diversification in Himalopsyche martynovi Species Complex (Trichoptera: Rhyacophilidae) in the Hengduan Mountains"

_biology, 2021, doi:10.3390/biology10080816_

Round 1

Reviewer 1 Report

Reviewer comment

I read the MS entitled “Gene flow and diversification in Himalopsyche martynovi species complex (Trichoptera: Rhyacophilidae) in the Hengduan Mountains” by Deng et al. This paper examines the morphological similarity, diffeence and variation among the clades of tHimalopsyche martynovi species complex distributed in the Hengduan Mountains, one of the biodiversity hotspots in the world. Authors appear to success to approach their goal well with detailed gene flow analyses based on a large number of nuclear loci data. The extent to which morphological variation caused by such high gene flow is a general phenomenon is questionable. In addition, it would be desirable to examine in more detail whether the observed gene flow and morphological changes are indeed correlated. However, there are few examples of studies using molecular markers of morphological changes caused by crossbreeding and gene flow, and I personally think that this study is worthy of publication with minor modifications.

Suggested modifications are listed below.

Line 89: monophyletic clade > a clade or monophylyetic group

Clade contains the meaning monophyly. Correct the same writings throughout the MS.

Line 264: Himalopsyche. cf. (Italic) > Himalopsyche cf. (Non-Italic)

Delete dot after Himalopsyche and non-Italicize cf.

Fig.2. In A and B, the tree topologies of species and RaxML trees are the same for the OTUs 3-5. Correct the torsions of RaxML tree topologies for OTUs 3-4 in A and B.

Fig.4. If possible, please include a photo of the entire individual not only for H. epikur but also all clades.

Line 327- 330: The sentence “The sample location map (Fig. 1) showed that H. epikur was distributed in south but H. martynovi was distributed more north. The distribution area of H. viteceki and the outgroup H. immodesta overlap with that of H. epikur.” would be nice to moved to M&M section.

Line 418: Delete “We believe”.

Author Response

Response to Reviewer 1:

Thank you very much for reviewing our manuscript and giving very useful comments which help to improve the quality of this manuscript a lot, we appreciate for your time and patient!

For this general comment: The extent to which morphological variation caused by such high gene flow is a general phenomenon is questionable. In addition, it would be desirable to examine in more detail whether the observed gene flow and morphological changes are indeed correlated. However, there are few examples of studies using molecular markers of morphological changes caused by crossbreeding and gene flow, and I personally think that this study is worthy of publication with minor modifications.

We acknowledge and agree that we cannot make a general statement on how relevant gene flow or introgression might be on generating intermediate morphologies. In our manuscript we thus hypothesize that gene flow may be the source of morphological variations in caddisflies. But indeed, it would be desirable to examine the correlation between gene flow and morphological condition under controlled conditions. This is, however, only possible in a cross-breeding study on these species, which for logistic and biological reasons is very difficult with these species and far beyond the scope of this study.

For the other suggestions, we have carefully corrected the manuscript based on each comment, and the point-by-point answers are listed below:

Comment 1: Line 89: monophyletic clade > a clade or monophylyetic group

Clade contains the meaning monophyly. Correct the same writings throughout the MS.

Response: Thank you for pointing out this inconsistency. We have modified this accordingly throughout the manuscript.

Comment 2: Line 264: Himalopsyche. cf. (Italic) > Himalopsyche cf. (Non-Italic)

Delete dot after Himalopsyche and non-Italicize cf.

Response: We have changed this as suggested.

Comment 3: Fig.2. In A and B, the tree topologies of species and RaxML trees are the same for the OTUs 3-5. Correct the torsions of RaxML tree topologies for OTUs 3-4 in A and B.

Response: We have edited the figure as suggested and updated it in the manuscript.

Comment 4: Fig.4. If possible, please include a photo of the entire individual not only for H. epikur but also all clades.

Response: Sadly, it is not possible to provide full habit images for the other species used in this study, as the studied specimens were all dissected for the study. The image of H. epikur was taken prior to dissection, identification and DNA extraction. Hence, we can only show a full habit image of H. epikur as an example.

Comment 5: Line 327- 330: The sentence “The sample location map (Fig. 1) showed that H. epikur was distributed in south but H. martynovi was distributed more north. The distribution area of H. viteceki and the outgroup H. immodesta overlap with that of H. epikur.” would be nice to moved to M&M section.

Response: We have moved this sentence from the Results section to the M&M section.

Comment 6: Line 418: Delete “We believe”.

Response: We have deleted the words as suggested.

Reviewer 2 Report

The authors performed integrative taxonomy and phylogenetic analyses in Himalopsyche martynovi species complex using AHE data and morphological characters, and they discussed the gene flow which might influence the diversification in the Hengduan Mts. I think the manuscript is well written with sufficient evidences and interesting results, which could be published as it is. Although the sampling seem to be still limited especially for this species with only one specimen. 

Some minor concerns:

I think you could think of modifying a bit in the abstract to avoid dividing it into different the sections.

Lack of space between numbers and bp in some places like line 236, 238, and etc. in the text.

In line 185, I wonder have you ever compared the results using GTR+I+G in your analysis, and is there any difference?

Reviewer 3 Report

The work investigate the diversification of the caddisflies Himalopsyche martynovi species complex, endemic from Hengduan Mountains (China). The work used an integrative approach combine classic morphological characters with genomic analysis of 989 loci. Phylogenetic and network analysis sheds light on the taxonomic identification of the species complex.

The work done is notably and with high value. I support it for the publication in the journal Biology. I suggested only minor elements to modify.

Throughout the text the wording sensu stricto have been used, and specified each time in the captions. For a better reading, I suggest to explain it the first time in the main text and use the abbreviation s.s. throughout the work. Please correct “sensu strict” in all captions.

Line 135. Could you please provide a very short description of library preparation and sequencing method used? E.g. the name of the kit used and sequencing technology used, for a more quick and usable consultation.

Line 304. Please specified also in the caption the meaning of the numbers value inside the arrows of the gene flow.

Line 479. Use the initial name for the author contributions

Line 487. Reference. Please standardize the names of the journals, abbreviations for all or not according to the request of the journal and use the same font for all (pay attention to capital letters). Enter the DOI for all or not according to the request of the journal.

Line 533. There is an error at the end of the citation, please correct it and insert the pages number.

Line 555 and Line 619. Please correct the journal name written in capital letters

Line 577. Please correct the citation and insert the name of the journal

Line 615. Please correct the journal name (GSA Bulletin)

Author Response

Response to Reviewer 3:

Thank you very much for reviewing our manuscript and giving helpful suggestions. We appreciate for your time and effort! All the suggestions are very useful for improving the quality of this manuscript, we have carefully corrected the manuscript based on each suggestion, and the point-by-point answers are listed below:

Point 1: Throughout the text the wording sensu stricto have been used, and specified each time in the captions. For a better reading, I suggest to explain it the first time in the main text and use the abbreviation s.s. throughout the work. Please correct “sensu strict” in all captions.

Response 1: Thank you for these suggestions, we have used the abbreviation s.s. instead of sensu stricto throughout the manuscript and corrected the captions.

Point 2: Line 135. Could you please provide a very short description of library preparation and sequencing method used? E.g. the name of the kit used and sequencing technology used, for a more quick and usable consultation.

Response 2: We have added more details about the library preparation and sequencing method in the Materials and methods (Line 137) section as suggested.

Point 3: Line 304. Please specified also in the caption the meaning of the numbers value inside the arrows of the gene flow.

Response 3: We have specified the meaning of the numbers value inside the arrows in the caption of figure 4 as suggested.

Point 4: Line 479. Use the initial name for the author contributions

Response 4: We changed corrected it as suggested.

Point 5: Line 487. Reference. Please standardize the names of the journals, abbreviations for all or not according to the request of the journal and use the same font for all (pay attention to capital letters). Enter the DOI for all or not according to the request of the journal.

Line 533. There is an error at the end of the citation, please correct it and insert the pages number.

Line 555 and Line 619. Please correct the journal name written in capital letters

Line 577. Please correct the citation and insert the name of the journal

Line 615. Please correct the journal name (GSA Bulletin)

Response 5: Thank you very much for your detailed suggestions! We have carefully checked and corrected all the references based on your comments and the journal guidelines.